# Enhancement of the Detection Performance of Paper-Based Analytical Devices by Nanomaterials

**DOI:** 10.3390/molecules27020508

**Published:** 2022-01-14

**Authors:** Renzhu Pang, Qunyan Zhu, Jia Wei, Xianying Meng, Zhenxin Wang

**Affiliations:** 1Department of Thyroid Surgery, The First Hospital of Jilin University, Changchun 130021, China; pangrenzhu@jlu.edu.cn (R.P.); weijia@ciac.ac.cn (J.W.); 2State Key Laboratory of Electroanalytical Chemistry, Changchun Institute of Applied Chemistry, Chinese Academy of Sciences, Changchun 130022, China; zhuqy@ciac.ac.cn; 3School of Applied Chemical Engineering, University of Science and Technology of China, Hefei 230026, China

**Keywords:** paper-based analytical devices, nanomaterials, point-of-care testing, signal enhancement

## Abstract

Paper-based analytical devices (PADs), including lateral flow assays (LFAs), dipstick assays and microfluidic PADs (μPADs), have a great impact on the healthcare realm and environmental monitoring. This is especially evident in developing countries because PADs-based point-of-care testing (POCT) enables to rapidly determine various (bio)chemical analytes in a miniaturized, cost-effective and user-friendly manner. Low sensitivity and poor specificity are the main bottlenecks associated with PADs, which limit the entry of PADs into the real-life applications. The application of nanomaterials in PADs is showing great improvement in their detection performance in terms of sensitivity, selectivity and accuracy since the nanomaterials have unique physicochemical properties. In this review, the research progress on the nanomaterial-based PADs is summarized by highlighting representative recent publications. We mainly focus on the detection principles, the sensing mechanisms of how they work and applications in disease diagnosis, environmental monitoring and food safety management. In addition, the limitations and challenges associated with the development of nanomaterial-based PADs are discussed, and further directions in this research field are proposed.

## 1. Introduction

As an easily accessible and cheap material made from cellulose (the most abundant polymer on earth) or nitrocellulose, paper offers many advantages for development of biosensing platforms, in particular point-of-care-testing (POCT) devices [1,2,3,4,5,6,7]. For instance, various in situ analyses can be achieved by the paper-based analytical devices (PADs) because many recognition probes, such as ligands, antibodies and aptamers, can be easily immobilized within the (nitro)cellulose matrix [8,9,10,11,12,13,14,15,16,17,18,19,20]. Because of the controlled porosity and capillary forces of the nitrocellulose/cellulose network, the fluidics can be efficiently transported via capillary flow. In addition, the PADs are compatible with different detection systems, including naked eye and simple optical or electrical readers, which meets the needs in developing countries [21,22,23,24,25]. In the past decades, PADs, including lateral flow assays (LFAs), dipstick assays and microfluidic PADs (μPADs), have been well developed and shown a great impact on the clinical diagnosis, environmental monitoring and food safety management [8,9,10,11,12,13,14,15,16,17,18,19,20,21,22,23,24,25,26,27,28,29,30,31,32,33,34,35]. For example, one type of PAD, the lateral flow immunoassay (LFIA), is commercially available and is extensively used as a powerful diagnostic platform for the rapid detection of antibodies or antigens at a low cost [34,35,36]. The LFIA has dominated the market of rapid diagnostic testing since the lateral flow immunochromatographic strip was first developed for screening the supernatants of hybridomas in 1982 through antigen–antibody interaction on paper to produce a color change visible to the naked eye [34,35,36,37].

The PADs detection methods include optical techniques (colorimetry, fluorescence and surface enhancement Raman scattering (SERS), etc.) and electrochemical (EC) methods (amperometry, potentiometry, voltammetry, electrochemical impedance spectroscopy (EIS), electrochemiluminescence (ECL) and photoelectrochemistry (PEC)) [21,22,23,24,25]. Among these detection methods, colorimetry offers simplicity and convenience and, until 2009, had been one of the main detection methods in PADs. The biggest advantage of paper-based colorimetric devices is that the presence of a specific analyte can be distinguished easily with the naked eye without expensive and complex instruments through the change of color. However, the colorimetric method is limited to qualitative yes/no detections and/or semi-quantitative analysis because it has several inherent disadvantages, such as narrow dynamic range, poor sensitivity and being easily interfered with by environmental light and biased by users’ subjectivity. The EC method was first used as a PAD detection technique in 2009 by Dungachi et al. [38]. Generally, the analytical performances (especially sensitivity) of electrochemical paper-based analytical devices (also known as ePADs) are better than those of paper-based colorimetric analytical devices [18,23,24,25,28,31]. Unfortunately, the analytical performance (in particular, selectivity) of ePADs can be decreased significantly in complex matrices. Currently, the EC detection and optical detection are accounted for as the two main detection methods of PADs (as shown in Figure 1).

To resolve these limitations, nanomaterials were utilized to produce selective and sensitive detection signals on PADs because nanomaterials and their composites exhibit unique physical and chemical properties (as shown in Figure 2) [26,27,28,29,30,31,32,33,34,35]. For instance, nanoparticles, including gold nanoparticles (AuNPs) and magnetic nanoparticles (MNPs), have been extensively used as signal indicators for paper-based colorimetric analytical devices [32,33,34]. Several AuNPs labeled lateral-flow test-strip (LFTS) immunosensors, such as human chorionic gonadotropin (HCG) and Hepatitis B surface antigen (HBsAg) colloidal gold immunoassay strips, have been clinically approved for rapid testing. Transition metal dichalcogenides (TMDs) and carbon nanomaterials, such as carbon nanotubes and graphenes, can accelerate electron transfer and increase actual electrode area when they are used as functional materials on electrode surfaces, resulting in the enhancement of the sensitivities of ePADs [18,24,31]. Fluorescent nanoparticles, such as quantum dots (QDs), carbon nanodots (CDs) and upconversion nanoparticles (UCNPs), offer new opportunities to update the assaying performance of paper-based fluorescent analytical devices in the responding time, sensitivity and selectivity because they have unique optical properties, such as tunable fluorescence color, high quantum yields, wide excitation wavelength, narrow emission band and excellent optical stability [33]. Moreover, nanomaterials have large amounts of surface-active sites as well as high surface-to-volume ratios, which support diverse functionalization with high densities of recognition units. The phenomenon further improves the detection performances of PADs. Therefore, the integration of nanomaterials with PADs enables strong quantitative capabilities of PADs and expands their applicable fields.

Currently, great efforts are being made for improving the detection performance of PADs by using advanced materials, such as nanomaterials and their composites [33,34,35,36]. The latest reviews have extensively summarized the specific characteristics of PADs. For instance, the fabrication of nanomaterial-based colorimetric and fluorescent PADs was reviewed by Patel et al. [33]. The signal amplification strategies of nanoparticle-based lateral flow testing strips (LFTSs) have been discussed by Shirshahi and Liu [34], and Díaz-González and de la Escosura-Muñiz [35], respectively. The engineering strategies for enhancing the performance of ePAD were reviewed by Baharfar et al. [39]. Based on the detection targets, the applications of nanomaterial-based PADs have also been reviewed by several groups [27,28,36]. The purpose of this review is to introduce readers to a general overview of the recent developments regarding nanomaterial-based PADs in terms of the detection modes and their representative applications.

## 2. Nanomaterial-Enhanced Paper-Based Analytical Device

### 2.1. Electrochemical Paper-Based Analytical Devices

Due to its desirable features, such as high sensitivity, rapid response and easy miniaturization, the EC detection has gradually become one of the most commonly used detection principles for PADs. A typical immunoassay of ePAD is shown in Figure 3. After adding a drop of analyte solution on the sample zone, the analyte will bind to the detection antibody at the conjugate zone and then bind to the capture antibody at the test zone; the excess conjugates are migrated to the absorbent zone under driving by capillary action. Three electrodes (working electrode, counter electrode and reference electrode) are needed for EC analysis, and the concentration of the target analyte can be correlated to the EC response intensity of the electroactive species. The electroactive species are produced by labels catalyzed EC substrates. Dungachi et al. fabricated the first paper-based ePAD for the simultaneous determination of glucose and lactate in real samples by photolithography and screen-printing technology in 2009 [38]. Various nanomaterials and their nanocomposites have been demonstrated as powerful EC transducers and efficient electroactive label carriers in the design strategy of ePADs, which offers great improvements of the analytical performance of ePADs through increasing EC signal (e.g., current) production and decreasing background noise (i.e., enhancing signal-to-noise ratio (S/N)) (as shown in in Table 1) [40,41,42,43,44,45,46,47,48,49,50,51,52,53,54,55,56,57,58,59,60,61,62,63,64,65,66,67,68,69,70,71,72,73,74,75,76,77].

Various nanoparticles, such as noble metal nanoparticles, metallic oxide nanoparticles and silica nanoparticles (SiNPs), have been extensively used to fabricate and/or modify the working electrodes of ePADs for achieving good detection performance through different methods, including directly dispersing nanoparticles in the printing ink and in situ growth, electrogeneration and drop-casting of nanoparticles on the screen-printed carbon electrodes (SPCE) [40,41,42,43,44,45,46,47,48,49,50,51,52,53,54,55,56]. For example, Pavithra et al. developed an ePAD for immunosensing carcinoembryonic antigen (CEA) by using AuNP working electrode, which was fabricated by directly screening printed self-made AuNP ink on the Whatman^®^ grade 1 chromatography paper [47]. The as-developed ePAD exhibited a linear range of 1.0 ng mL^−1^ to 100.0 ng mL^−1^ with a limit of detection (LOD) of 0.33 ng mL^−1^, The ePAD was used successfully to analyze CEA in the diluted human serum samples, demonstrating that the ePAD has good practicability. Zheng et al. developed an ePAD for ultrasensitive detection of CEA and prostate-specific antigen (PSA) by using cyclodextrin functionalized AuNPs (CD@AuNPs) and AuNPs modified paper working electrode (PWE) [40]. The CD@AuNPs exhibited mimicking properties of both glucose oxidase (GOX) and horseradish peroxidase (HRP) simultaneously, which can efficiently electrocatalytically reduce H_2_O_2_. The AuNPs modified PWE was constructed by in situ growth of AuNPs on the surfaces of cellulose fibers of paper. Taking advantage of the high conductivity of AuNP modified PWE and good catalytic activity of CD@AuNPs, the as-developed ePAD exhibited wide linear ranges (0.005 to 100 ng mL^−1^ (CEA) and 0.002 to 40 ng mL^−1^ (PSA)), low LODs (0.002 ng mL^−1^ (CEA) 0.001 ng mL^−1^ (PSA) and high stability (retaining 90% of the initial responses after stored at 4 °C for 15 days). The ePAD was used to detect CEA and PSA in spiked human serum samples, and satisfactory recoveries (in the range of 100.4% to 109.2% for CEA and in the range of 100.9% to 114.0% for PSA) were obtained, indicating that the ePAD has a great potential application of detecting CEA and PSA in clinical samples. Pinyorospathum et al. developed an ePAD for detection of C-reactive protein (CRP) through electrodeposited AuNPs on SPCE, followed by the self-assembly of PMPC-SH on AuNP surface [41]. In the presence of CRP and Ca^2+^, the current of ePAD was decreased by increasing the concentration of CRP, while [Fe(CN)_6_]^3−/4−^ was used as the electrochemical probe. The as-developed ePAD exhibited a linear range of 5 to 5000 ng·mL^−1^ with an LOD of 1.6 ng·mL^−1^, which was applied successfully to detect CRP in the certified human serum samples. De França developed an ePAD for detection of dopamine through drop-casting CdSe/CdS magic-sized QDs (MSQDs) on the graphite working electrode (5 mm in diameter), which was drawn on chromatography paper by 6B grade pencil [52]. After being decorated by 10 μg MSQDs, the peak current was enhanced ca. 46% in comparison with that of bare electrode, besides a decrease in the charge transfer resistance and increase in the electroactive area of the sensor. The as-developed ePAD exhibited excellent analytical performance, including low LOD (96 nmol L^−1^), good stability (within a period of 7 days without major variations of peak current), repeatability (ca. 2.85% relative standard deviation (RSD) and reproducibility (ca. 7.2% RSD). The ePAD was employed successfully for sensing dopamine in human blood serum samples with recovery rates between 95.2% and 102.6%. Because of their high catalytic activity, nanomaterials can also be used as active probe for developed ECL PAD [54,55,56]. For instance, Huang et al. developed an auto-cleaning ECL PAD for detection of Ni^2+^ and Hg^2+^ through using superior peroxidase-like activity of cubic Cu_2_O-Au nanoparticles and a large specific surface area and excellent conductivity of silver nanoparticles (AgNPs) [55]. The cubic Cu_2_O-Au nanoparticles can catalyze H_2_O_2_ to generate reactive oxygen species, promoting the luminescence of N-(4-Aminobutyl)-N-ethylisoluminol (ABEI)). The as-developed ECL PAD exhibited wide linear ranges (10 nmol L^−1^ to 0.2 mmol L^−1^ (Ni^2+^) and 10 pmol L^−1^ to 1 μmol L^−1^ (Hg^2+^)) and low LODs (3.1 nmol L^−1^ (Ni^2+^) and 3.8 pmol L^−1^ (Hg^2+^)), which was used successfully to detect Ni^2+^ and Hg^2+^ in the spiked lake water.

As transduction materials, carbon nanomaterials, including single-walled carbon nanotubes (SWCNTs), multi-walled carbon nanotubes (MWCNTs) and different types of graphene materials, have been getting great attention in the ePAD fabrication because they have high surface area, excellent electrical conductivity and rich surface-chemical properties [57,58,59,60,61,62,63,64,65,66,67,68,69,70,71,72,73]. For instance, Valentine et al. found that the device-to-device reproducibility and current intensity of ePAD can be efficiently improved through formation of MWCNT network in the porous structures of paper [58]. Tran et al. developed an ePAD for non-enzymatic detection of glucose through the deposition of the SWCNT layer on wax-printed nitrocellulose (NC) membrane [59]. The SWCNT electrode exhibits high conductivity with an average resistivity of less than 100 Ω/sq and excellent mechanical property. After modification of the SWCNT electrode by AuNPs, the as-developed ePAD exhibited excellent glucose detection performance, including good reproducibility (RSD < 8%) and high sensitivity (240 μA/mM cm^2^), which was used successfully to determine glucose in Coke. Pungjunun et al. developed an origami-based ePAD for sensing NO and NO_2_ (as NO_X_) by using a screen-printed graphene electrode modified with copper nanoparticles (CuNP/SPGE) [64]. Because of the good catalytic property of CuNP towards the EC conversion of NO_X_ and excellent conductivity of graphene, the as-developed ePAD exhibited high selectivity, low LODs (0.23 vppm and 0.03 vppm with exposure times of 25 min and 1 h, respectively), good reproducibility (RSD < 5.1%) and long lifetime (>30 days). The ePAD was applied to detect NO_X_ in air and exhaust gases from cars, and satisfactory results were obtained. Cai group has been developed a series of ePADs for detection of various biomarkers, including CEA and neuronspecific enolase (NSE), by using amino functional graphene (NG)-Thionin (THI)-AuNPs nanocomposites modified SPCEs [66,67]. Integration of SiNPs modified paper microzones with reduced graphene (RG) modified SPCE, Scala-Benuzzi et al. developed an ePAD for the quantitative determination of ethinylestradiol (EE2) in water samples through capturing EE2 by the immobilized anti-EE2 specific antibodies on the paper microzones, subsequently releasing the adsorbed EE2 by adding a diluted solution of sulfuric acid and detecting the desorbed EE2 by Osteryoung square wave voltammetry (OSWV) [69]. The as-developed ePAD exhibited excellent analytical performance, including wide linear range (0.5 to 120 ng L^−1^), low LOD (0.1 ng L^−1^) good recovery values (from 97% to 104%) and good reproducibility (RSD < 4.9%). There are no significant differences were found between the results of ePAD and the results of spectrophotometric immunoassay, when two methods were used for the quantification of EE2 in river water samples and spiked water samples.

The unique features of metal-organic frameworks (MOFs), including high porosity, tunable framework structures, large surface areas and multiple functionalities, make them extremely attractive for improving the detection performance of biosensors [78,79]. Recently, MOFs and their nanocomposites have been used for developing ePAD with excellent detection performance [75,76,77]. Wei et al. fabricated a cobalt-MOF (Co-MOF) modified carbon cloth/paper (CC/Paper) hybrid button-based PAD (Co-MOF/CC PAD) for nonenzymatic quantitative EC detection of glucose through in situ growth of Co-MOF on CC [75]. As a typical nanozyme, the environment tolerance of Co-MOFs is much better than that of natural enzyme, which can increase significantly the stability of ePAD. Densely grown Co-MOF on CC can maximize its catalytic sites, resulting in high sensitivity of ePAD. The Co-MOF/CC PAD exhibits linear range from 0.8 mmol L^−1^ to 16 mmol L^−1^ with an LOD of 0.15 mmol L^−1^ and maintains at a stable detection performance in 60 days, and then gradually decreased to about 60% after 120 days. The ePAD was used successfully to determine glucose in multiple body fluids, including serum, urine and saliva. Lu et al. developed an ePAD-based DNA hybridization for detection of human immunodeficiency virus (HIV) DNA by using the nickel MOF (Ni-MOF) composite/AuNPs/CNTs/polyvinyl alcohol (Ni–Au composite/CNT/PVA) paper electrode as working electrode and methylene blue (MB) as a redox indicator [77]. The CNT/PVA were deposited on the cellulose membrane by vacuum filtration, and Ni–Au composites were loaded on CNT/PVA film by the drop-casting method. The Ni–Au composite/CNT/PVA film electrode has large specific surface area and conjugated π-electron system, which makes a higher loading of the single-stranded DNA probe than that of CNT/PVA film electrode. The phenomenon improves the sensitivity for detecting target DNA. The ePAD exhibited excellent sensing performance with a wide linear range of 10 nmol L^−1^ to 1 μmol L^−1^, a low LOD of 0.13 nmol L^−1^, good selectivity against one-base mismatch DNA sequences and excellent stability after 20 days of storage. The target HIV DNA was detected successfully even in complex serum samples by the as-developed ePAD.

### 2.2. Colorimetric Paper-Based Analytical Devices

Colorimetric detection is the most common method in PADs. Figure 4 shows a typical schematic of colorimetric test strip. Generally, the test strip includes five functional zones: sample, conjugate, test, control and absorbent zones. Under the analyte solution migration, conjugates will capture the analyte with detection antibody at conjugate zone, and then the conjugates and analyte bind to the capture antibody at test zone, the excess conjugates are migrated to the control zone conjugate with secondary antibody. After finishing the reaction, the ImageJ is usually used for the collecting colorimetric signals of the test and control lines. Up to now, various kinds of nanomaterials are used as colorimetric labels. The nanomaterial-based colorimetric PADs have been extensively used for detection of various targets (as shown in Table 2) [43,80,81,82,83,84,85,86,87,88,89,90,91,92,93,94,95,96,97,98,99,100,101,102,103,104,105,106,107,108,109,110,111,112]. Plasmonic nanoparticles, such as AuNPs and AgNPs, are extremely useful indicators for the fabrication of colorimetric PADs because of their strong local surface plasmon resonance (SPR) bands [80]. Moreover, multiple, simultaneous tests can be rapidly performed with low sample consumption by incorporating these surface-modified AuNPs into a PADs that can be read using just a smartphone. For example, Díaz-Amaya et al. developed a μPAD for multiplexed aptamer-based detection of analytical targets through a salt-induced aggregation of single strand DNA (ssDNA) functionalized polyethyleneimine (PEI) encapsulation of gold-decorated polystyrene (PS) core particles (ssDNA-PEI-Au-PS) [85]. The net positive charge of the PEI layer avoids the direct interaction of metallic ions and ssDNAs with the AuNPs, which provides ideal conditions for controlled induction of aggregation of AuNPs on the test zones of μPAD, resulting in high sensitivity, selectivity and reproducibility (RSD = 5.69%). Using a smartphone as detector, the analytical performance of as-proposed μPAD was demonstrated by multiplexed detection of Hg^2+^ and As^3+^ with low LODs (1 μg mL^−1^) and high specificity (*p* > 0.05) versus different interferent ions (Ca^2+^, Fe^2+^, Mg^2+^ and Pb^2+^). The authors provided a universal idea for fabricating μPAD with the capability of multiplex and quantitative colorimetric detection. Monisha et al. reported a PAD with AgNPs for on-site determination of Hg^2+^ from environmental water samples by inkjet-printing polyvinyl pyrrolidine (PVP) stabilized AgNPs on Whatman^®^ grade 1 chromatography paper [96]. In the presence of Hg^2+^, the color of AgNP is changed from yellow to colorless because of the oxidation of AgNPs into Ag^+^ ions on the paper substrate. The as-proposed PAD exhibited the linear range from 40 to 1200 ng mL^−1^ with LOD of 10 ng mL^−1^, and was used successfully for quantitative detection of Hg^2+^ in different types of water samples collected from river, tube well, pond, coal mines and industrial waste. This approach could be used to determine Hg^2+^ in other real samples, such as biological and vegetable samples. Recently, Mettakoonpitak et al. developed an uncomplicated, affordable and environmentally friendly method for fabrication of μPAD by screen-printing biodegradable polycaprolactone (PCL) as high-resolution hydrophobic barriers [98]. The proposed method can produce as narrow as 510 ± 40 μm hydrophilic channel and 490 ± 30 μm hydrophobic edge, respectively. The as-developed method was used successfully to fabricate μPADs for detection of Cr^3+^ and Cl^−^ with high selectivity. For Cr^3+^ analysis, the μPADs achieved a linear range of 50.0 to 1000.0 ng mL^−1^ with an LOD of 15.0 ng mL^−1^, when AgNPs were used as the colorimetric probe. Integration of nanoparticle as color indicator and PCL screen-printing could provide a simple and environmentally friendly method for fabricating μPADs with high analytical performance, which are ultimately utilized in wide-ranging applications.

In addition, the AuNPs can also serve as carriers for the simultaneous immobilization of different biomolecules (e.g., antibodies, DNA and aptamers) with an abundant number of biorecognition elements or optical/electrochemical tags on their surfaces to provide more binding sites or signal amplification of analyte for a single recognition reaction. Furthermore, different antibodies can be easily introduced to the AuNP surface via electrostatic interactions to provide highly specific recognition sites for biomolecular sensing, resulting in simplify the PAD fabrication procedure. Huang et al. reported a highly sensitive colorimetric PAD for detection of PSA by using AuNPs labeled with biotinylated poly(adenine) ssDNA sequences and streptavidin-HRP for enzymatic signal enhancement [87]. The as-proposed PAD was able to detect as low as 10 pg mL^−1^ PSA in a test that could be completed in as little as 15 min.

Comparison with natural enzymes, nanomaterials with enzyme-like characteristics (i.e., nanozymes), such as magnetic nanoparticles, noble metal nanoparticles, MOFs, heterojunctions, etc., exhibit several advantages, including easy production with large-scale, low cost and high stability in harsh environments. These unique properties endow them with attractive applications in the fabrication of PADs with high analytical performance [104,111,112]. Zhang et al. developed a ready-to-use PAD for the determination of H_2_O_2_ by simply immobilization of mesoporous carbon-dispersed palladium nanoparticles (Pd NPs/meso-C) and the 3,3′,5,5′-tetramethylbenzidine (TMB) substrate onto a common chromatography paper [104]. Taking the advantage of large surface area of the meso-C support and the good dispersity of PdNPs, the PdNPs/meso-C show excellent catalytic performance to trigger the chromogenic reaction of colorless TMB to blue TMBox mediated by H_2_O_2_. The as-developed PAD exhibited a linear range of 5 to 300 mol L^−1^, and can be used to determine H_2_O_2_ in complex matrices, such as milk. Kitchawengkul et al. developed a laminated three-dimensional (3D)-μPAD for colorimetric determination of total cholesterol (TC) in human blood by using the peroxidase-like activity of nitrogen-doped CDs (N-CDs) [108]. The 3D-μPAD with a 6 mm circular detection zone was fabricated by a simple wax screen-printing technique, which consisted of four layers laminated together vertically. The 3D-μPAD exhibited a linear range of 0.05 to 10 mmol L^−1^ with an LOD of 0.014 mmol L^−1^. In particular, TC in human blood could be determined by the naked eye within 10 min by simple comparison with a color chart. Overall, the as-proposed 3D-μPAD serves as a simple, low cost, rapid, sensitive and selective alternative for detection of TC in whole blood samples that is friendly to unskilled end users. Cui et al. fabricated an origami PAD (oPAD) assisted by Pd decorated Cu/Co co-doped CeO_2_ (CuCo-CeO_2_-Pd) nanospheres, for dual-mode electrochemical/visual detection of amyloid-β (Aβ) peptide with high sensitivity [48]. In this case, the CuCo-CeO_2_-Pd nanospheres were introduced as an enhanced “signal transducer layer”, which act as an outstanding catalyst for catalyzing glucose to produce H_2_O_2_ for DPV signal readout and further 3,3′,5,5′-tetramethylbenzidine (TMB) oxidation for colorimetric analysis. The oPAD exhibited linear ranges from 1.0 pmol L^−1^ to 100 nmol L^−1^ (EC detection) and 10 pmol L^−1^ to 100 nmol L^−1^ (visual detection) with LODs of 0.05 pmol L^−1^ (EC detection) and 0.5 pmol L^−1^ (visual detection), respectively. The oPAD was used successfully to analysis Aβ peptide in artificial cerebrospinal fluid (aCSF) and serum samples. Al Lawati et al. developed a PAD for the colorimetric/fluorometric monitoring of glucose by co-immobilizing two-dimensional cobalt-terephthalate MOF nanosheets (2D CoMOFs) and GOX on chromatography paper [112]. Due to its highly porous and extraordinarily stable structures, the 2D CoMOF increased significantly the stability and performance of GOX, and also acted as a catalyst to accelerate the reaction of H_2_O_2_ produced by the enzymatic oxidation of glucose with o-phenylenediamine (OPD) serving as a peroxidase substrate, resulting in a yellow-brown color change and a high fluorescence emission. The as-developed PAD showed high analytical performance for the quantification of glucose including high accuracy, wide linear range (50 mol L^−1^ to 15 mmol L^−1^) and low LODs (16.3 (colorimetric detection) and 3.2 mmol L^−1^ (fluorometric detection)), and was used successfully to determine glucose in blood samples from healthy and diabetic volunteers.

### 2.3. Fluorometric Paper-Based Analytical Devices

Recently, the nanomaterial-based fluorometric PADs are increasingly being developed for sensing various targets (as shown in Table 3) [113,114,115,116,117,118,119,120,121,122,123,124,125,126,127,128,129,130,131,132,133,134,135,136,137,138,139,140,141,142,143,144,145,146,147,148,149,150,151,152,153,154,155,156,157,158,159,160,161]. The schematic diagram of the working principle of the fluorometric PAD is shown in Figure 5. The reaction process of the analyte on the test strip is common with that of the colorimetric test strip. After the reaction is finished, the fluorometric signals of emission wavelengths at the test and control zones are recorded by fluorometric spectrophotometer under irradiating with excitation light. Fluorescent nanomaterials, including metal nanoclusters (NCs), CDs, QDs, UCNPs and MOFs, have unique properties, such as wide excitation wavelength, narrow emission band, tunable fluorescence color, highly optical stability and good surface-modified flexibility. For instance, Ungor et al. developed a fluorescent PAD with red-emitting (λ_emission_ = 645 nm, d = 1.5 ± 0.3 nm) fluorescent gold nanoclusters (AuNCs) for rapid detection of L-kynurenine (Kyn) with an LOD of 5 μmol L^−1^, which is in good accordance with the toxic Kyn concentration of liquor and serum for several cancers (vulvar, ovarian cancer and leukemia) [116]. Lert-itthiporn et al. reported a fluorescent PAD for membraneless gas-separation with subsequent determination of iodate (IO_3_^−^) by fabrication of two circular reservoirs (donor reservoir and the acceptor reservoir) in a folded chromatography paper [117]. The IO_3_^−^ is reduced to free iodine (I) by iodide (I^−^) in the donor reservoir, while the gold core of the bovine serum albumin-stabilized gold NCs (BSA-AuNCs) in the acceptor reservoir is etched by diffused I from the donor reservoir, which results in the quenching of the red emission of BSA-AuNCs in the acceptor reservoir. After folding, the donor reservoir and acceptor reservoir were mounted together through a two-sided mounting tape to allow membraneless gas-separation of free I from the donor reservoir to diffuse into the acceptor reservoir. Under the ultraviolet (UV) light (365 nm) irradiation, the PAD exhibited a linear range from 0.005 to 0.1 mmol L^−1^, an LOD of 0.01 mmol L^−1^, high accuracy (mean recovery: 95.1 (±4.6) %) and high precision (RSD < 3%), which was applied successfully for the measurement of IO_3_^−^ in iodized salts and fish sauces without prior sample pre-treatment. Yin et al. developed a fluorescent/colorimetric dual-model PAD based on the quenching effect of graphitic carbon nitride on palladium nanoclusters (PdNCs) for detection of miRNA let-7a [119]. In this case, the PdNCs not only was used as a fluorescence probe but also could catalyze a chromogenic reaction for the generation of color change. Combined with nucleic acid cycle signal amplification, the fluorescent/colorimetric dual-model PAD exhibited linear ranges of 50 pmol L^−1^ to 1 mol L^−1^ (colorimetry) and 10 fmol L^−1^ to 1 nmol L^−1^ with LODs 16 pmol L^−1^ (colorimetry) and 3 fmol L^−1^ (fluorescence), respectively. In addition, the fluorescent/colorimetric PAD has excellent stability (about 90% of the fluorescent response remaining after 6 weeks) and good reproducibility (both of intra-assay and inter-assay RSDs were less than 5.5%). The experimental results demonstrate that the fluorescent/colorimetric dual-model PAD can be used for on-site detection of miRNAs with good analytical performance.

QDs are generally good donors for the fabrication of FRET sensing platforms since the fluorescence of QDs is easily quenched by many substances. Liu et al. reported a fluorescent PAD for detection of Ag^+^ and AgNPs by inkjet-printing CdTe QDs on Whatman^®^ grade 3030–861 chromatography paper because Ag^+^ enables to quench the fluorescence of CdTe QDs via a cation exchange reaction between Ag^+^ and the CdTe QDs [121]. Under optimized conditions, the as-proposed PAD exhibited high analytical performance of Ag^+^ or AgNPs (pretreated by HNO_3_), including high selectivity, low LOD (0.05 μg mL^−1^) and good accuracy (4.5% and 2.2% RSDs for 1 μg mL^−1^ and 7 μg mL^−1^ of Ag^+^, respectively). In addition, the practicality of the fluorescent PAD was demonstrated by detecting Ag^+^ and AgNPs in river water and 12 commercial products, including four textiles, three gynecologicallotions, one surgical dressing and four baby products. Zhou et al. developed a 3D rotary fluorescent PAD for multiplexed detection of Cd^2+^ and Pb^2+^ by transferring the liquid phase of ZnSe QDs@ion imprinted polymers to solid glass fiber paper [132]. Under optimized experiment conditions, the as-proposed 3D rotary fluorescent PAD exhibited a linear range from 1 to 70 ng mL^−1^ with an LOD of 0.245 ng mL^−1^ for Cd^2+^, and a linear range from 1 to 60 ng mL^−1^ with an LOD of 0.335 ng mL^−1^ for Pb^2+^, respectively. Qu et al. developed a fluorescent PAD based on polythiophene (CP)-CdTe QD conjugates for detection of acetylcholinesterase (AChE) by turning on the fluorescence of the CP-CdTe QD conjugates via the interaction between CP and thiocholine produced by ATCh hydrolysis and aggregation induced emission enhancement (AIEE) [125]. Under optimized conditions, the as-developed fluorescent PAD exhibited a low LOD of 0.14 U L^−1^. Zhang et al. developed a fluorescent PAD for the ratiometric fluorescence determination of 2,4-dichlorophenoxyacetic acid through fluorescence resonance energy transfer (FRET) of nitrobenzoxadiazole (NBD) and CdTe QDs [122]. Under optimized conditions, the as-developed fluorescent PAD exhibited a linear range of 0.56 to 80 μmol L^−1^ with an LOD of 90 nmol L^−1^. The PAD was applied successfully for detection of 2,4-dichlorophenoxyacetic acid in spiked soybean sprouts and lake water samples with high recovery rates ranging from 86.2% to 109.5% and the RSD less than 4.19%.

As a new class of fluorescent nanomaterials, CDs (also known as carbon QDs and carbon dots) have been used to fabricate fluorescent PADs because of their superior merits, such as easy synthesis, biocompatibility, environmental friendliness, fluorescence tunability and stable luminescent emission [141,142,143,144,145,146,147,148,149,150,151,152]. Wang et al. developed an instrument-free fluorescent PAD for detection of Pb^2+^ by directly inject-printing dual-emission CDs (blue CDs and red CDs) in A4 paper [144]. The blue fluorescence was quenched by Pb^2+^, while the red fluorescence was kept unchanged. The as-developed fluorescent PAD can detect as low as 2.89 nmol L^−1^ Pb^2+^ under a 365 nm UV lamp irradiation. The fluorescent PAD was used successfully to determine Pb^2+^ in real samples, including tape water and lake water. Tian et al. developed a fluorescent PAD for detection of F^−^ in water by immobilizing the Ca^2+^, CDs and hexametaphosphate capped AuNPs on the cellulose chromatography paper [149]. Under a 365 nm UV lamp irradiation, the fluorescence color of the PAD changed from orange to blue through the aggregation induced FRET mechanism when various concentrations of F^−^ (0, 100, 200, 300 and 400 mol L^−1^) were applied. Li et al. developed a fluorescent PAD based on hybrid polydimethylsiloxane (PDMS)/paper for detection of folic acid (FA) by using CDs as fluorophores, which were immobilized on the cellulose paper by Schiff base chemistry [143]. Under optimized conditions, the as-developed fluorescent PAD exhibited a wide range of 1 to 300 μmol L^−1^ with an LOD of 0.28 μmol L^−1^. The feasibility of the fluorescent PAD was further verified by detection of FA in orange juice and urine samples with satisfactory results. Liang et al. developed a flower-like AgNPs (FLS)-enhanced fluorescent/visual bimodal PAD for detection of multiple miRNAs [151]. In this case, the ssDNA functionalized CDs (DNA1-N-CDs) were immobilized on the FLS layer, which was in situ grown on the surfaces of cellulose fibers of chromatography paper. The fluorescences of DNA1-N-CDs were quenched by ssDNA functionalized CeO_2_ NPs (DNA2-CeO_2_) through DNA hybridization. In the presence of miRNA, the fluorescent intensities of DNA1-N-CDs were recovered and strengthened by FLS. The disengaged DNA2-CeO_2_ could result in color change after adding H_2_O_2_, leading to the real-time visual detection of miRNA. The as-developed FLS-enhanced fluorescent PAD exhibited linear ranges of 0.1 fmol L^−1^ to 1 nmol L^−1^ and 0.2 fmol L^−1^ to 2 nmol L^−1^ for miRNA210 and miRNA21 with LODs of 0.03 fmol L^−1^ for miRNA210 and 0.06 fmol L^−1^ for miRNA21, respectively. The practicability of the FLS-enhanced fluorescence PAD was demonstrated by successful detection of miRNA210 in different cell lysates.

Because of NIR-excitation and the visible light emission nature of UCNPs, the fluorescent PADs using UCNPs as the label can avoid the interference of autofluorescence and scattering light from biological samples and paper substrates, resulting in an improvement in the detection accuracy of the PAD [153,154,155]. Recently, He et al. developed a UCNP-based fluorescent PAD for detection of total immunoglobulin E (IgE) in human serum through resonance energy transfer between UCNPs and organic dye tetramethylrhodamine (TAMRA) [155]. The UCNP-based fluorescent PAD exhibited a linear range of 0.5 to 50 IU mL^−1^ with an LOD of 0.13 IU mL^−1^. The practicability of the UCNP-based fluorescent PAD was demonstrated by the determination of IgE in 20 human serum samples. The results of the UCNP-based PAD were well consistent with those of the commercial ELISA kit. The RSDs (n = 3) of the PAD varied from 2.7% to 19.7%. The results suggested that the UCNP-based fluorescent PAD could be used as a POCT device for individual diagnostic and real-time detection.

Recently, a MOFs-based fluorescent PAD has been developed for the detection of various targets [157,158,159,160,161]. Lv et al. developed a fluorescent PAD for detection of CEA through wet NH_3_-triggered structural change of NH_2_-MIL-125(Ti) impregnated on paper [157]. The NH_2_-MIL-125(Ti)-based PAD exhibited a linear range of 0.1 ng mL^−1^ to 200 ng mL^−1^ with an LOD of 0.041 ng mL^−1^. Yue et al. developed a portable smartphone-assisted ratiometric fluorescent PAD for detection of malachite green (MG) by using fluorescent Al-MOF nanosheet and rhodamine B (RhB) as fluorescent probes [161]. The as-developed fluorescent PAD exhibited a wide linear range of 0.5 to 200 μg mL^−1^, a low LOD of 1.6 μg mL^−1^, satisfactory recoveries (in the range of 81.90% to 108.00% and low RSD (in the range of 1.00% to 4.69%). The practicability of the fluorescent PAD was verified by detection of MG in spiked fish tissues. The as-obtained results were in good agreement with those obtained by high performance liquid chromatography (HPLC).

### 2.4. Paper-Based Surface-Enhanced Raman Spectroscopic Analytical Devices

The basic principle of SERS is that the signal of the analyte is strongly amplified through LSPR phenomena (i.e., electromagnetic hot spots) generated by light when it interacts with labels (plasmonic metal nanoparticles), such as gold nanorods (GNRs) and AgNPs, as shown in Figure 6. The PAD-based SERS substrates have gained considerable attention since they enable on-site label-free detection of a wide variety of analytes and provide “fingerprint” signatures of analytes (as shown in Table 4) [134,162,163,164,165,166,167,168,169,170,171,172,173,174,175,176]. Saha and Jana developed a PAD-based SERS assay for the detection of proteins by mixing plasmonic nanomaterials (silver coated AuNPs (Ag@AuNPs)) and analyte in the mobile phase, where the analyte induced Ag@Au nanoparticles form controlled aggregates and generate electromagnetic hot spots inside the microfluidic channel, resulting in a strong SERS signal [173]. The as-developed PAD-based SERS assay exhibited high reproducibility and sensitivity, which can be used to detect 1 fmol L^−1^ concanavalin A within 3 min. Qi et al. developed an oPAD for miRNA detection through modification of DNA-encoded Raman-active anisotropic AgNPs in the hydrophilic channels [171]. In the presence of analyte, the Raman signals on DNA-encoded AgNPs were amplified through a target-dependent, sequence-specific DNA hybridization assembly. The simple and low-cost oPAD is generic and applicable to various miRNAs, which holds promising applications in point-of-care diagnostics because it can be used to detect as low as 1 pmol L^−1^ within 15 min. Wu et al. developed a PAD for detection of acrylamide (AAm) in food products by using the strawberry-like SiO_2_/Ag nanocomposites (SANC) immersed chromatography paper [172]. Under the optimized conditions, the as-developed PAD SERS assay exhibited a wide linear response from 0.1 nmol L^−1^ to 50 μmol L^−1^ with a low LOD of 0.02 nmol L^−1^ and good recoveries of 80.5% to 105.6% for practical samples, including cookies, chips and bread. In addition, the total assaying time of the PAD was less than 10 min. The result indicated that the PAD SERS assay could be a promising strategy in food analysis and verification. Li et al. developed a colorimetric/SERS dual-mode PAD for sensing SO_2_ by immobilization of 4-mercaptopyridine (Mpy)-modified GNRs-reduced graphene oxide (rGO) hybrids (rGO/MPy-GNRs), anhydrous methanol and starch-iodine complex into cellulose-based chromatography paper through a vacuum filtration method [174]. The PAD can be used not only as a naked-eye indicator of SO_2_ changed from blue to colorless but also as a highly sensitive SERS substrate because of the SO_2_-triggered conversion of Mpy to pyridine methyl sulfate on the GNRs. The PAD-based SERS method exhibited a wide linear range from 1 μmol L^−1^ to 2000 μmol L^−1^ with a low LOD of 1 μmol L^−1^. The colorimetric/SERS method was employed for the detection of SO_2_ in wine, and the as-obtained results matched well with those obtained from the traditional Monier-Williams method. In addition, the color intensities and profiles of the SERS spectra of the colorimetric/SERS dual-mode PAD after 10 weeks are very similar to those of freshly prepared PADs, indicating excellent stability of the colorimetric/SERS dual-mode PAD. This study provides a new strategy for designing of paper-based sensing platform for a wide range of on-site testing applications. Moreover, taking advantage of different nanomaterials, Xia et al. developed a smart PAD (named as vapor generation (paper-based thin-film microextraction system) capable of both sensitive on-site fluorescence detection and accurate SERS quantification of volatile benzaldehyde (BA) by utilizing stimuli-responsive core@shell GNR@QD-embedded MOF structures [125]. The fluorescence emission of carboxyl-capped QDs was completely quenched by amino-modified GNRs via electrostatic interaction. In the presence of BA, the GNRs-QD assemblies was dissociated through the Schiff base reactions between the amine group of 4-mercaptonoaniline and the aldehyde moiety of BA, resulting in the increase in the fluorescence and Raman signal of the hybrid systems. In addition, gaseous BA molecules can be efficiently and selectively concentrated on the GNR surface through the “cavity-diffusion” effect of porous MOF shells, allowing the discrimination of BA in exhaled breath rapidly and precisely even at the sub-ng mL^−1^ level with excellent specificity against other volatile organic compounds. The as-developed fluorescent/SERS dual-mode PAD was used successfully to accurately discriminate lung cancer from controls.

### 2.5. Comparison of the Detection Methods of Paper-Based Analytical Devices

The above introduced detection methods of PADs indicate that their detection performances have been greatly enhanced with the great advance of nanofabrication science. For the convenience of the readers, the merits and drawbacks of the four major PAD detection methods (EC, colorimetric, fluorometric, SERS PADs) are compared and summarized in Table 5.

## 3. Conclusions and Perspective

The PADs have gained remarkable consideration as simple, low-cost and powerful POCT platforms since the first commercialized lateral flow immunoassay (LFIA) was used for a home pregnancy test. Although PADs have achieved great success in the rapid testing area, some parameters on the analytical performance of PADs need to be further improved to meet the specific needs of different detection fields. Integrating nanomaterials/nanotechnology into PADs can help improve their analytical performance, including sensitivity, selectivity, reproducibility, stability and multiplexed analysis. For instance, the sensitivity of ePADs can be increased significantly when carbon nanomaterials with high conductivity and high specific surface area are used as electrode materials or electrode modifiers. The AuNPs are commonly used color indicators of colorimetric PADs. The sensitivity of colorimetric PADs will be further increased when AuNPs are conjugated with other catalytic NPs, such as CeO_2_ NPs, and/or enlarged through the metallic atoms (such as Au or Ag) surface deposition strategy. In comparison with organic dyes, the fluorescent nanomaterials, including QDs, CDs and metallic NCs, have wide excitation wavelength, high fluorescence brightness and strong resistance to photobleaching, resulting in the excellent stability, reliability and accuracy of PADs. Because the upconverted phosphorescence of UCNP can efficiently avoid the interference of biological autofluorescence, the reliability and accuracy of PADs can also be improved by using UCNP as a fluorescence probe. Nanozymes such as CeO_2_ NPs and MOFs have higher robustness than natural enzymes. Therefore, the PADs exhibit high stability and sensitivity, while the nanozymes are used for signal amplification in the PAD fabrication. Meanwhile, the nanomaterials have surface-functionalized flexibility for creating multiple reactive/recognizable sites with analytes to achieve multiplexed analysis. In addition, the nanomaterial-based PADs are compatible with dual-mode detection, such as colorimetry–fluorescence and EC-optical detection, which enables the reliability and accuracy of the PADs.

The nanomaterial-based PADs have managed great achievements for sensing various targets, including ions, small molecules, nucleic acids, proteins and pathogens, in the bench research. However, there are limited examples that have transformed from proof-of-principle analytical devices to commercial products. More efforts should be made continuously to develop novel strategies for increasing the longevity, robustness and reliability in the real-time monitoring, which are largely dependent on the synthesis of advanced nanomaterials, introduction of a new sensing mechanism and multiple detection modes, development of a reliable microfabricating methodology/standard and simplifying the detection procedures. The next generation of PADs will be simple, cost-effective and multiplexed and be able to provide on-site qualitative/quantitative analysis by the naked eye and/or portable equipment, such as smartphones and wearable electronic devices.

## Figures and Tables

**Figure 1 molecules-27-00508-f001:**
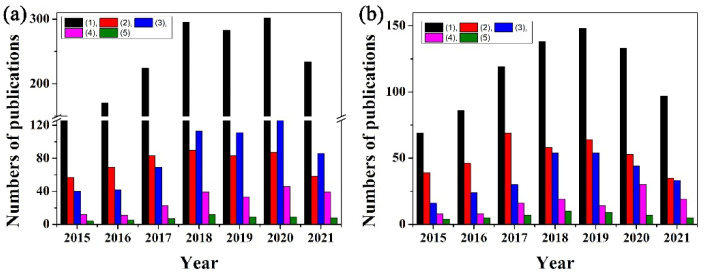
The annual publications numbers from 1 January 2015 to 30 October 2021 on the keywords: (**a**) (1) paper-based analytical devices and paper-based analytical devices plus (2) electrochemical, (3) colorimetric, (4) fluorescence, (5) Raman; and (**b**) (1) paper-based analytical devices plus nanoparticle/nanoparticles, and paper-based analytical devices plus nanoparticle or nanomaterial plus (2) electrochemical, (3) colorimetric, (4) fluorescence, (5) Raman. The data were obtained from Web of Science^TM^.

**Figure 2 molecules-27-00508-f002:**
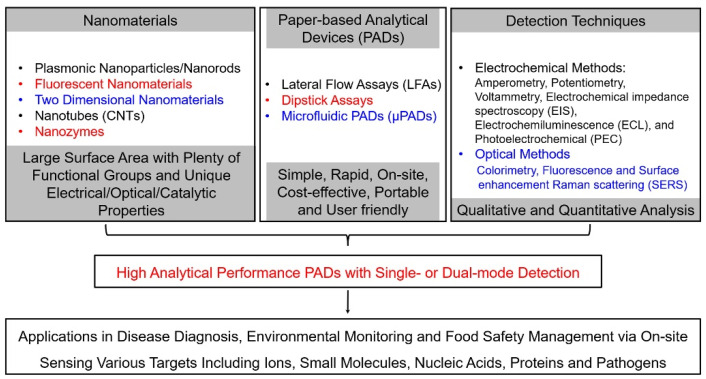
Outline of nanomaterial-based paper-based analytical devices.

**Figure 3 molecules-27-00508-f003:**
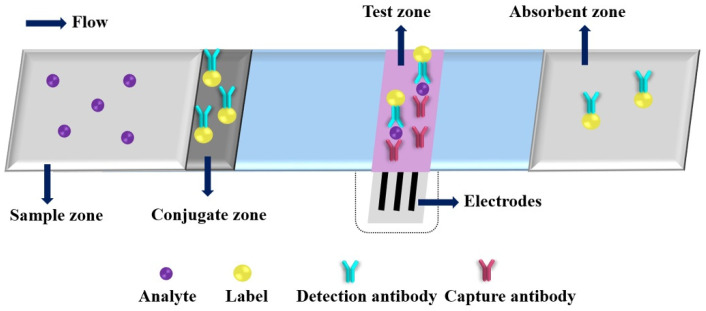
An illustration of a typical electrochemical paper-based analytical device.

**Figure 4 molecules-27-00508-f004:**
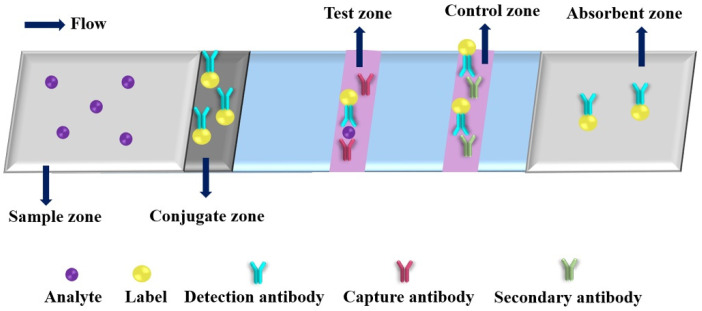
An illustration of a typical colorimetric paper-based analytical device.

**Figure 5 molecules-27-00508-f005:**
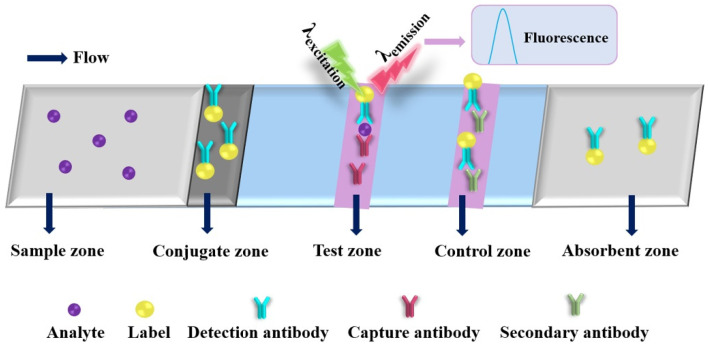
Schematic diagram of a typical fluorometric paper-based analytical device.

**Figure 6 molecules-27-00508-f006:**
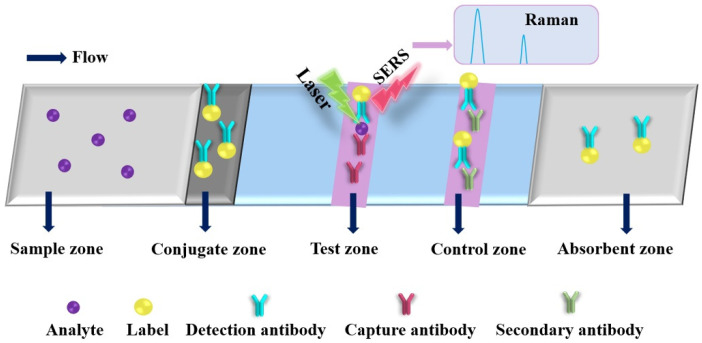
Schematic diagram of a typical paper-based surface-enhanced Raman spectroscopic analytical device.

**Table 1 molecules-27-00508-t001:** The typical nanomaterial-enhanced ePADs for sensing various analytes.

Nanomaterials	Modification Methods	Electrochemical Method	Analytes	Linear Ranges	Limit of Detection	Real Samples	Recovery	Ref.
AuNPs	In situ growth	DPV	CEA and PSA	5 × 10^−3^ to 100 ng mL^−1^ (CEA) and 2 × 10^−3^ to 40 ng mL^−1^ (PSA)	2 × 10^−3^ ng mL^−1^ (CEA) 1 × 10^−3^ ng mL^−1^ (PSA)	Human serum	-	[40]
AuNPs	Electrodeposition	DPV	CRP	5 to 5 × 10^3^ ng mL^−1^	1.6 ng mL^−1^	Certified human serum	-	[41]
AuNPs	Electrodeposition	DPV	EGFR	0.5 to 500 nmol L^−1^	0.167 nmol L^−1^	Saliva samples	-	[42]
AuNPs	Drop-casting	DPV	H1047R (A3140G) missense mutation in exon 20	-	5 nmol L^−1^ (signal on) and 6 nmol L^−1^ (signal off)	-	-	[43]
AuNPs	Electrodeposition	Impedimetry	miRNA 155	0 to 4 × 10^3^ ng mL^−1^	6.9 × 10^2^ ng mL^−1^ (93.4 nmol L^−1^)	Fetal bovine serum	-	[44]
AuNPs	Drop-casting	SWASV	Hg^2+^	5 to 200 ng mL^−1^	2.5 ng mL^−1^	Drinking water	95% to 104%	[45]
Poly (N-vinylpyrolidone) AuNPs	Screen-printing	Chronoamperometry	Glucose	1 × 10^4^ to 1.5 × 10^6^ nmol L^−1^	2.6 × 10^4^ nmol L^−1^	-	-	[46]
Poly (N-vinylpyrolidone) AuNPs	Screen-printing	DPV	CEA	1 to 100 ng mL^−1^	0.33 ng mL^−1^	Diluted human serum	99.58% to 102.50%	[47]
Pd decoration of Cu/Co-doped CeO_2_ (CuCo-CeO_2_-Pd) nanospheres and urchin-like AuNPs	In situ growth	DPV	Amyloid-β	1 × 10^−3^ to 100 nmol L^−1^	5 × 10^−5^ nmol L^−1^	Artificial cerebrospinal fluid and human serum	99% to 100.5%	[48]
N-CDs, TiO_2_ NPs and Pt NPs	Drop-casting	PEC	CEA	2 × 10^−3^ to 200 ng mL^−1^	1.0 × 10^−3^ ng mL^−1^	Living MCF-7 cells	-	[49]
TiO_2_ nanosheets and CeO_2_ NPs	In situ growth of TiO_2_ nanosheets and drop-casting of CeO_2_ NPs	PEC	Thrombin	2 × 10^−5^ to 0.1 nmol L^−1^	6.7 × 10^−6^ nmol L^−1^	Human serum	-	[50]
CdSe/CdS magic-sized QDs	Drop-casting	DPV	Dopamine	500 to 1.5 × 10^4^ nmol L^−1^	96 nmol L^−1^	Human serum	95.2% to 102.6%	[51]
ZnO NPs	Drop-casting	SWV	Picric acid	4 × 10^3^ to 6 × 10^4^ nmol L^−1^	4.04 × 10^3^ nmol L^−1^	Tap water, lake water	92.3% to 98.9%	[52]
Molecularly imprinted polymer coated Fe_3_O_4_@Au@SiO_2_ NPs	Drop-casting	LSV	Serotonin	10 to 10^6^ nmol L^−1^	2 nmol L^−1^	Pharmaceutical capsules and urine samples	100% to 111%	[53]
Patchy gold coated Fe_3_O_4_ nanospheres	-	ECL	CEA	1 × 10^−4^ ng mL^−1^ to 15 ng mL^−1^	3 × 10^−5^ ng mL^−1^	Human serum	-	[54]
Cubic Cu_2_O-Au NPs and AgNPs	In situ growth (AgNPs)	ECL	Ni^2+^ and Hg^2+^	10 to 2 × 10^5^ nmol L^−1^ (Ni^2+^) and 1 × 10^−2^ nmol L^−1^ to 1 × 10^3^ nmol L^−1^ (Hg^2+^)	3.1 nmol L^−1^ (Ni^2+^) and 3.8 × 10^−3^ nmol L^−1^ (Hg^2+^)	Lake water	96.4% to 101.6% (Ni^2+^) and 96.0% to 104.0% (Hg^2+^)	[55]
DNA-functionalized PtCu nanoframes	-	ECL	Streptavidin	1 × 10^−4^ nmol L^−1^ to 100 nmol L^−1^	3.34 × 10^−5^ nmol L^−1^	Human serum	98.4% to 106.5%	[56]
SWCNTs	Vaccum filtration	CV	Glucose	5 × 10^−5^ to 1 × 10^7^ nmol L^−1^	1.48 × 10^5^ nmol L^−1^	Coke	97.3% to 105%	[59]
Graphene	Screen-printing	DPV	Oxytetracycline	1 to 200 ng mL^−1^	0.33 ng mL^−1^	Milk, honey and shrimp	-	[60]
Graphene	Screen-printing	DPV	Hepatitis B virus DNA	5 × 10^−2^ to 100 nmol L^−1^	1.45 × 10^−3^ nmol L^−1^	Plasmid constructs	-	[61]
Graphene	Screen-printing	SWV	Salivary thiocyanate	2.5 × 10^4^ to 7 × 10^5^ nmol L^−1^	6 × 10^3^ nmol L^−1^	Human saliva	-	[62]
Graphene	Drop-casting	DPV	ATP	3 × 10^2^ to 4.5 × 10^5^ nmol L^−1^	80 nmol L^−1^	Human serum and cell lysates	95.4% to 104.2%	[63]
Graphene and CuNPs	Screen-printing (graphene) and in situ growth CuNPs	DPV	NOx	0 to 150 vppm	0.23 vppm	Ambient indoor and outdoor air, and exhaust gases	-	[64]
Graphene and AuNPs	Drop-casting (graphene) and label (AuNPs)	DPV	Pseudopodium-enriched atypical kinase one	1 × 10^−2^ to 1 × 10^3^ ng mL^−1^	1 × 10^−2^ ng mL^−1^	Human serum	103% to 104%	[65]
(NH_2_-G)/Thi/AuNPs nanocomposites	Drop-casting	DPV	CEA	5 × 10^−2^ to 500 ng mL^−1^	1 × 10^−2^ ng mL^−1^	Human serum	-	[66]
(NH_2_-G)/Thi/AuNPs nanocomposites	Drop-casting	DPV	CEA and NSE	1 × 10^−2^ to 500 ng mL^−1^ for CEA and 5 × 10^−2^ to 500 ng mL^−1^ for NSE	2 × 10^−3^ ng mL^−1^ for CEA and 1 × 10^−2^ ng mL^−1^ for NSE	Human serum	-	[67]
Reduced graphene	Screen-printing and in situ EC reduction	Amperometry	Claudin 7 and CD81	2 × 10^−3^ to 1 ng mL^−1^ (Claudin 7) and 0.01 to 10 ng mL^−1^ (CD81)	4 × 10^−4^ ng mL^−1^ (Claudin 7) and 3 × 10^−3^ ng mL^−1^ (CD81)	Plasma of breast cancer patients	-	[68]
Reduced graphene	Drop-casting and EC reduction	SWV	Ethinylestradiol	5 × 10^−4^ to 0.12 ng mL^−1^	1 × 10^−4^ ng mL^−1^	River water	97.5% to 103.7%	[69]
Graphene QDs	Drop-casting	SWV	UA and creatinine	10 to 3 × 10^3^ nmol L^−1^	8.4 nmol L^−1^ (UA) and 3.7 nmol L^−1^ (Creatinine)	Human urine	98.9% to 101.5%	[70]
GO	Drop-casting	SWV	CRP, cTnI and PCT	1 to 1 × 10^5^ ng mL^−1^ (CRP), 1 × 10^−3^ to 250 ng mL^−1^ (cTnI), 5 × 10^−4^ ng mL^−1^ to 250 ng mL^−1^ (PCT)	0.38 ng mL^−1^ (CRP), 1.6 × 10^−4^ ng mL^−1^ (cTnI) and 2.7 × 10^−4^ ng mL^−1^ (PCT)	Human serum	-	[71]
RGO and cysteine AuNPs	Drop-casting (RGO) and electrodeposition (AuNPs)	Chronoamperomatriy	IL-8	1 × 10^−3^ to 9 × 10^−3^ ng mL^−1^	5.89 × 10^−4^ ng mL^−1^	-	-	[72]
(NH_2_-GO)/Thi/AuNPs	Screen-printing	DPV	EGFR	5 × 10^−2^ to 200 ng mL^−1^	5 × 10^−2^ ng mL^−1^	Human serum	-	[73]
rGO/Thi/S-AuNP/Chi	Drop-casting	Amperometry	17β-E2	1 × 10^−2^ to 100 ng mL^−1^	1 × 10^−2^ ng mL^−1^	Human serum	-	[74]
Cobalt-MOF	In situ growth	Amperometry	Glucose	8 × 10^5^ to 1.6 × 10^7^ nmol L^−1^	1.5 × 10^5^ nmol L^−1^	Serum, Urine and Saliva	87.2% to 108.6%	[75]
Pd@hollow Zn/Co core−shell ZIF67/ZIF8 NPs	Drop-casting	DPV	PSA	5 × 10^−3^ ng mL^−1^ to 50 ng mL^−1^	7.8 × 10^−4^ ng mL^−1^	-	-	[76]
Ni-MOFs/AuNPs/MWCNTs/PVA	Vacuum filtration (MWCNT/PVA) and drop-casting (Ni-MOFs/AuNPs)	DPV	HIV DNA	10 to 1 × 10^3^ nmol L^−1^	0.13 nmol L^−1^	Human serum	95.5% to 103.8%	[77]

CV: cyclic voltammetry; SWV: square wave voltammetry; ECL: electrochemical luminescence; PEC: photoelectrochemical; DPV: differential pulse voltammetry; EIS: electrochemical impedance spectroscopy; ASV: anodic stripping voltammetry; LSV: linear-sweep voltammetry; SWASV: square wave anodic stripping voltammetry; 2D: two dimension; 3D: three dimension; NPs: nanoparticles; AuNPs: gold nanoparticles: CDs: carbon dots; N-CDs: nitrogen-doped carbon dots; QDs: quantum dots; AgNPs: silver nanoparticles; SWCNTs: single-walled carbon nanotubes; MWCNTs: multiple-walled carbon nanotubes; MOFs: metal-organic frameworks; Ni-MOFs: nickel metal-organic frameworks; CuNPs: copper nanoparticles; (NH_2_-G)/Thi/AuNPs: amino functionalized graphene/thionine/gold nanoparticles nanocomposites; RGO: reduced graphene oxide; GO: graphene oxide; rGO/Thi/S-AuNP/Chi: amino redox graphene/thionine/streptavidin-modified gold nanoparticles/chitosan; ZIF: zeolite imidazole ester framework material; PVA: polyvinyl alcohol; CEA: carcinoembryonic antigen; PSA: prostate-specific antigen; CRP: C-reactive protein; EGFR: epidermal growth factor receptor; NSE: neuronspecific enolase; ATP: adenosine triphosphate; UA: uric acid; IL-8: interleukin 8; cTnI: cardiac troponin I; PCT: procalcitonin.

**Table 2 molecules-27-00508-t002:** The typical nanomaterial-enhanced colorimetric PADs for sensing various analytes.

Nanomaterials	Analytes	Linear Ranges	Limit of Detection	Real Samples	Recovery	Ref.
AuNPs	Gallic acid	1 × 10^4^ to 1 × 10^6^ nmol L^−1^	1 × 10^3^ nmol L^−1^	Tea	85.2% to 93.1%	[80]
Avidin functionalized AuNPs	Ig G	-	300 ng mL^−1^	-	-	[81]
Citrate stabilized AuNPs	Melamine	100 to 10^6^ ng mL^−1^	100 ng mL^−1^	Milk	-	[82]
Citrate stabilized AuNPs	NADH	-	1.25 × 10^4^ nmol L^−1^	Cell Lysate	-	[83]
Aspartic acid modified AuNPs	Cysteine	9.99 × 10^4^ to 9.987 × 10^5^ nmol L^−1^	1 × 10^3^ nmol L^−1^	Human plasma	99.2% to 101.1%	[84]
ssDNA-PEI-Au-PS	Hg^2+^ and As^3+^	0 to 3 × 10^4^ ng mL^−1^	1 × 10^3^ ng mL^−1^	River water	96.2% to 116.7%	[85]
ssDNA functionalized AuNPs	Tuberculosis DNA	1.95 × 10^−2^ to 19.5 ng mL^−1^	1.95 × 10^−2^ ng mL^−1^	Infected tissue	-	[86]
ssDNA functionalized AuNPs	PSA	-	1 × 10^−2^ ng mL^−1^	Human serum	-	[87]
Antibody functionalized AuNPs	Ig G	-	284.52 ng mL^−1^	Whole blood	-	[88]
Antibody functionalized AuNPs	Yersinia Pestis	-	2.5 × 10^−2^ ng mL^−1^	-	-	[89]
Antibody functionalized AuNPs	Influenza A H1N1 and H3N2 viruses	-	2.7 × 10^3^ pfu/assay for H1N1 detection and 2.7 ×10^4^ pfu/assay for H3N2 detection	Cell lysate	-	[90]
Antibodies functionalized AuNRs	sIL-2R	1 to 6.25 × 10^3^ ng mL^−1^	1.0 ng mL^−1^	Mouse serum	93% to 109%	[91]
Antibodies functionalized AuNRs	CRP	50 to 1 × 10^4^ ng mL^−1^	1.3 ng mL^−1^	Human plasma	-	[92]
Co(II) catalyst, secondary antibody, luminol multifunctionalized AuNPs	H-FABP, cTnI and copeptin	1 × 10^−4^ to 1 × 10^3^ ng mL^−1^, 5 × 10^−4^ to 1 × 10^3^ ng mL^−1^ and 1 × 10^−3^ to 1 × 10^6^ ng mL^−1^ for H-FABP, cTnI and copeptin	6 × 10^−5^ ng mL^−1^, 3 × 10^−4^ ng mL^−1^ and 4 × 10^−4^ ng mL^−1^ for H-FABP, cTnI and copeptin	Human serum	94% to 108%	[93]
Cu/Co-doped CeO_2_ (CuCo-CeO_2_-Pd) nanospheres and urchin-like AuNPs	Amyloid-β	1 × 10^−2^ to 100 nmol L^−1^	5 × 10^−4^ nmol L^−1^	Artificial cerebrospinal fluid (aCSF) and human serum	99% to 100.5%	[48]
Gold nanostars	Glucose	0 to 2 × 10^7^ nmol L^−1^	1.4 × 10^6^ nmol L^−1^	-	-	[94]
PVP stabilized AgNPs	Nitrite	10 to 5 × 10^3^ nmol L^−1^ and 10^4^ to 3.2 × 10^6^ nmol L^−1^	8.5 × 10^−2^ nmol L^−1^	Tap, river and lake water	95.6% to 101.9%	[95]
PVP stabilized AgNPs	Hg^2+^	40 to 1.2 × 10^3^ ng mL^−1^	10 ng mL^−1^	Tube well, river pond water, industrial waste and coal mine water	92.5% to 96.0%	[96]
PVP stabilized AgNPs	Ascorbic acid	1 × 10^6^ to 4 × 10^6^ nmol L^−1^	8.28 × 10^4^ nmol L^−1^	Vitamin C tablet and artificial juice	-	[97]
Citrate stabilized AgNPs	Cr^3+^ and Cl^−^	50 to 1 × 10^3^ ng mL^−1^ (Cr^3+^) and 1 × 10^4^ to 5 × 10^5^ ng mL^−1^ (Cl^−^)	15 ng mL^−1^ (Cr^3+^) and 1 × 10^4^ ng mL^−1^ (Cl^−^)	Instant noodle seasoning	-	[98]
Citrate stabilized AgNPs	Hg^2+^	1 to 4 ng mL^−1^	0.86 ng mL^−1^	River water	98.9% to 101%	[99]
Achillea Wilhelmsii extract coated AgNPs	Hg^2+^	1 × 10^3^ to 7 × 10^5^ nmol L^−1^	300 nmol L^−1^	River, well and lake water	-	[100]
Citrate capped Cu@Ag core@shell NPs	Phenthoate	50 to 1.5 × 10^3^ ng mL^−1^	15 ng mL^−1^	Pond and river water, cucumber and potato	92.6% to 97.4%	[101]
DNA-templated Ag/Pt NCs	miRNA21	1 × 10^−3^ to 0.7 nmol L^−1^	6 × 10^−4^ nmol L^−1^	Human urine	93.8% to 106.0%	[102]
ssDNA functionalized PtNPs	miRNAs	1 × 10^−2^ to 100 nmol L^−1^	8.5 × 10^−3^ nmol L^−1^ (miRNA21)	Human Serum	86.2% to 112.2%	[103]
Pd NPs/meso-C	H_2_O_2_	5 × 10^3^ to 3 × 10^5^ nmol L^−1^	1 × 10^3^ nmol L^−1^	Commercial milk	100.9% to 109.7%	[104]
ZnONRs	Glucose and UA	Glucose (1 × 10^4^ to 1 × 10^7^ nmol L^−1^) and uric acid (1 × 10^4^ to 5 × 10^6^ nmol L^−1^)	3 × 10^3^ nmol L^−1^ for glucose and 4 × 10^3^ nmol L^−1^ for uric acid	Human serum and urine	89% to 109%	[105]
Cr_2_O_3_-TiO_2_ nanocomposites	H_2_O_2_	5 to 1 × 10^5^ nmol L^−1^	3 nmol L^−1^	Tap water, milk and fetal bovine serum (FBS) albumin	95.8% to 98%	[106]
Mn-ZnS QDs	Glyphosate	5 to 5 × 10^4^ ng mL^−1^	2 ng mL^−1^	Whole grain	80.6% to 119.9%	[107]
N-CDs	H_2_O_2_	5 × 10^4^ to 1 × 10^7^ nmol L^−1^	1.4 × 10^4^ nmol L^−1^	Human plasma	91.0% to 113.0%	[108]
CDs@Eu/GMP ICPs	Cerebral acetylcholinesterase	0.1 mU mL^−1^ to 60 mU mL^−1^	0.033 mU mL^−1^	Brain tissues and cerebral fluid	-	[109]
Carbon nitride nanoparticles	Tetracycline	800 to 4 × 10^5^ nmol L^−1^	120 nmol L^−1^	Shrimp samples and river water	98.7% to 102.8%	[110]
Poly(L-lactic acid) nanofibers	Glucose	1 × 10^5^ to 5 × 10^6^ ng mL^−1^	1 × 10^5^ ng mL^−1^	-	-	[111]
Cobalt (II)-terephthalate MOFs	Glucose	5 × 10^4^ to 1.5 × 10^7^ nmol L^−1^	3.2 × 10^3^ nmol L^−1^	Human blood	96.9% to 102.6%	[112]

ssDNA-PEI-Au-PS: single strand DNA (ssDNA) functionalized polyethyleneimine (PEI) encapsulation of gold-decorated polystyrene (PS) core particles; AgNPs: silver nanoparticles; PVP: poly(vinylpyrrolidone); NCs: nanoclusters; PdNPs: palladium nanoparticles; Pd NPs/meso-C: mesoporous carbon-dispersed PdNPs; PtNPs: platinum nanoparticles; AuNRs: gold nanorods; ZnONRs: zinc oxide nanorods; Ig G: immunoglobulin G; NADH: aihydronicotinamide adenine dinucleotide; H-FABP: heart-type fatty acid-binding protein; sIL-2R: soluble interleukin-2 receptor.

**Table 3 molecules-27-00508-t003:** The typical nanomaterial-enhanced fluorescent PADs for sensing various analytes.

Nanomaterials	Analytes	Linear Ranges	Limit of Detection	Real Samples	Recovery	Ref.
AuNCs/MIL-68(In)-NH_2_/Cys	Hg^2+^	0.02 nmol L^−1^ to 200 nmol L^−1^ and 200 to 6 × 10^4^ nmol L^−1^	6.7 × 10^−3^ nmol L^−1^	Tap and Lake water	91.3% to 110.2%	[115]
γG-AuNCs	L-kynurenine (Kyn)	-	5 × 10^3^ nmol L^−1^	Artificial cerebrospinal fluid	-	[116]
BSA-AuNCs	Iodate	5 × 10^3^ to 1 × 10^5^ nmol L^−1^	5 × 10^3^ nmol L^−1^	Iodized salts and fish sauces	90.5% to 102%	[117]
PVP-supported CuNCs	Iodine	100 to 500 ng mL^−1^	29 ng mL^−1^	-	97% to 108%	[118]
Graphitic carbon nitride nanosheets and ssDNA functionalized PdNCs	Let-7a	5 × 10^−2^ to 1 × 10^3^ nmol L^−1^ (Colorimetry) and 1 × 10^−5^ to 1 nmol L^−1^ (Fluorescence)	1.6 × 10^−2^ nmol L^−1^ (Colorimetry) and 3 × 10^−6^ nmol L^−1^ (Fluorescence)	Human serum	91% to 110%	[119]
Imprinted polymer grafted CdTe QDs	Cu^2+^, Cd^2+^, Pb^2+^ and Hg^2+^	-	10 ng mL^−1^ (Cu^2+^), 7 ng mL^−1^ (Cd^2+^), 9 ng mL^−1^ (Pb^2+^) and 15 ng mL^−1^ (Hg^2+^)	Seawater	-	[120]
CdTe QDs	Ag^+^ and Ag NPs	50 to 1.1 × 10^4^ ng mL^−1^ (Ag^+^)	50 ng mL^−1^ (Ag^+^)	River water and antibacterial Products	94% to 115%	[121]
CdTe QDs	2,4-dichlorophenoxyacetic acid	560 to 8 × 10^4^ nmol L^−1^	90 nmol L^−1^	soybean sprouts and lake water	86.2% to 109.5%	[122]
Mercaptosuccinic-acid capped CdTe QDs	Arsenic	50 to 3 × 10^4^ ng mL^−1^	16 ng mL^−1^	Water	92% to 112%	[123]
Silica-embedded CdTe QDs functionalized with rhodamine derivative	Fe^3+^	0 to 3.25 × 10^3^ nmol L^−1^	26.5 nmol L^−1^	Lake water and river water	94.2% to 106.0%	[124]
Polythiophene-coated CdTe QDs	Acetylcholinesterase	-	2.13 U L^−1^	Human serum	107% to 112%	[125]
Antibody functionalized CdTe QD and Au NPs	Immunoglobulin G	10 to 100 ng mL^−1^	0.4 ng mL^−1^	Human serum	97% to 104%	[126]
CdTe QDs and antibody functionalized AgNPs	Matrix metalloproteinase-7 (MMP7)	0.01 to 30 ng mL^−1^	7.3 × 10^−3^ ng mL^−1^	Human serum	91.7% to 113.3%	[127]
CdTe QDs embedded SiNPs	Alpha fetoprotein (AFP)	0.001 to 20 ng mL^−1^	400 ng mL^−1^	Human serum	-	[130]
CdTe/CdSe QDs	Carcinoembryonic antigen (CEA)	0.05 to 20 ng mL^−1^	6.7 × 10^−3^ ng mL^−1^	Human serum	-	[131]
ZnSe QDs	Cd^2+^ and Pb^2+^	1 to 70 ng mL^−1^ (Cd^2+^) and 1 to 60 ng mL^−1^ (Pb^2+^)	0.245 ng mL^−1^ (Ca^2+^) and 0.335 ng mL^−1^ (Pb^2+^)	Lake water and Seawater	95.0% to 105.1%	[132]
CdSexS_1_-x@ZnS (core@shell) QDs	Oligonucleotide biomarkers	-	1.5 × 10^−3^ nmol	-	-	[133]
GNR-QD core−shell embedded MOF structures	Benzaldehyde	2 to 5 × 10^3^ ng mL^−1^	1.2 ng mL^−1^	Human exhalation	-	[134]
ssDNA functionalized QDs coated MSNs and GO	MCF-7, HL-60 and K562 cells	180 to 8 × 10^7^ (MCF-7 cell), 210 to 7 × 10^7^ (HL-60 cell) and 200 to 7 × 10^7^ cells mL^−1^ (K562 cell)	62 (MCF-7 cell), 70 (HL-60 cell) and 65 (K562 cell) cells mL^−1^	-	-	[140]
N-CDs	Hg^2+^	1 × 10^4^ to 8 × 10^5^ nmol L^−1^	10.7 nmol L^−1^	-	-	[141]
N-CDs	Hg^2+^	500 to 2.5 × 10^4^ ng mL^−1^	500 ng mL^−1^	Drinking, pond and tap water	80% to 111%	[142]
CDs	Folic acid	1 × 10^3^ to 3 × 10^5^ nmol L^−1^	280 nmol L^−1^	Orange juice and urine	95.8% to 106.2%	[143]
Blue CDs and red CDs	Pb^2+^	0 to 200 nmol L^−1^	2.89 nmol L^−1^	Tap water and lake water	92.8% to115.2%	[144]
CDs and hexametaphosphate capped AuNPs	Ca^2+^, Mg^2+^ and F^−^	0 to 4.5 × 10^5^ nmol L^−1^ (F^−^)	2.1 × 10^4^ nmol L^−1^ (F^−^)	Ground water	96.2% to 109.5%	[149]
ssDNA functionalized N-CDs and ssDNA functionalized CeO_2_ NPs	miRNAs	1 × 10^−7^ to 1 nmol L^−1^ (miRNA210) and 2 × 10^−7^ to 2 nmol L^−1^ (miRNA21)	3 × 10^−8^ nmol L^−1^ (miRNA210) and 6 × 10^−8^ nmol L^−1^ (miRNA21)	Cell lysates	96.9% to 103.0%	[151]
CDs@Eu/GMP ICPs	Acetylcholinesterase	0 to 60 mU mL^−1^	2 mU mL^−1^	Brain tissues and cerebrospinal fluid (CSF)	98.3% to 98.8%	[152]
UCNPs	Immunoglobulin E (IgE)	0.5 to 50 IU mL^−1^	0.13 IU mL^−1^	Human serum	-	[155]
Eu@SiNPs	*Bacillus anthrax* spores	-	2.38 × 10^4^ spore mL^−1^	Yellow river water, tap water and soil	92.9% to 106.9%	[156]
NH_2_-MIL-125(Ti) MOF and GDH/antibody functionalized AuNPs	CEA	0.1 to 200 ng mL^−1^	0.041 ng mL^−1^	Human serum	-	[157]
Eu-DPA/PTA-NH_2_ MOF	H_2_O	0 to 100% *v/v*	0.01% *v/v*	Weisu granule, Cefuroxime axetil capsule and Azithromycin capsule	-	[158]
Cu(II)-Pyrazolate-based porphyrinic MOFs	Dopamine	2.5 to 1 × 10^3^ nmol L^−1^	2.5 nmol L^−1^	Human serum	-	[160]
Al-MOF nanosheet	Malachite green	500 to 2 × 10^5^ ng mL^−1^	1.6 × 10^3^ ng mL^−1^	Fish tissue	91.0% to 108.8%	[161]

AuNCs: gold nanoclusters; AuNCs/MIL-68(In)-NH_2_/Cys: glutathione stabilized AuNCs/indium-based MOFs modified with cysteine; BSA-AuNCs: bovine serum albumin-stabilized AuNCs; γG-AuNCs: γ-globulin (γG) immunoprotein stabilized AuNCs; SiNPs: silica nanoparticles; MSNs: mesoporous silica nanoparticles; UCNPs: upconversion nanoparticles; Eu@SiNPs: europium (Eu)-doped silicon nanoparticles; GDH: glutamate dehydrogenase; Eu-DPA/PTA-NH_2_ MOF: Eu-dipicolinic acid/2-aminophthalic acid MOF; ICP: ion concentration polarization.

**Table 4 molecules-27-00508-t004:** The typical nanomaterial-based SERS PADs for sensing various analytes.

Nanomaterials	Analytes	Linear Ranges	Limit of Detection	Real Samples	Recovery	Ref
GNR-QD core−shell embedded MOF structures	Benzaldehyde	0.1 to 10 ng mL^−1^	0.1 ng mL^−1^	Human exhalation	-	[134]
GNRs	Fluorescein and napthalenethiol	-	∼1 × 10^−7^ nmol L^−1^ (Fluorescein) and 5 × 10^−10^ nmol L^−1^ (Naphtalenethiol)	Tap water	-	[162]
GNRs modified with 4-mercaptophenylboronic acid (4-MBA) and 1-decanethiol (1-DT) molecules	Glucose	5 × 10^5^ to 1 × 10^7^ nmol L^−1^	1 × 10^5^ nmol L^−1^	Blood	88%	[163]
AuNPs	Cocaine	-	10 ng mL^−1^	Human plasma	-	[164]
AuNPs	Age-related macular degeneration aqueous humors: THV-I_1043_, THV-I_1454_ and THV-I_1656_	-	-	Aqueous humors	-	[165]
AuNPs	Clenbuterol	1 × 10^−4^ to 1 × 10^2^ ng mL^−1^	1 × 10^−4^ ng mL^−1^	Swine hair	104.8% to 116.2%	[166]
AgNPs	P-selectin	100 to 500 ng mL^−1^	104.2 ng mL^−1^ (ca. 0.7 nmol L^−1^)	-	-	[167]
Flower-like AgNPs	Chloramphenicol	1 × 10^−2^ to 1 × 10^5^ ng mL^−1^	1 × 10^−2^ ng mL^−1^	Pork	90% to 102%	[168]
Silver nanocubes	Adenine	10 to 1 × 10^5^ nmol L^−1^	0.89 nmol L^−1^	Urine	89% to 107%	[169]
4-aminothiophenol-modified rGO/ Ag NPs	Formaldehyde (FA) and acetaldehyde (AA)	4.5 × 10^−4^ to 480 ng mL^−1^	1.5 × 10^−4^ ng mL^−1^ (FA) and 1.3 × 10^−3^ ng mL^−1^ (AA)	Wine and human urine	104.6% to 112.8%	[170]
ssDNA functionalized anisotropic AgNPs	Has-miR-21	-	1 pmol L^−1^	-	-	[171]
Strawberry-like SiO_2_/Ag nanocomposites	Acrylamide	0.1 to 5 × 10^4^ nmol L^−1^	0.02 nmol L^−1^	Cookies, chips and bread	80.5% to 105.6%	[172]
Silver coated AuNPs functionalized with 4-Mercapto Pyridine and glucose or 4-mercapto pyridine and biotin.	Streptavidin and concanavalin A	-	1 × 10^−5^ nmol L^−1^ (Streptavidin) and 1 × 10^−6^ nmol L^−1^ (Concanavalin A)	-	-	[173]
4-mercaptopyridine (Mpy)-modified GNRs-rGO hybrids	SO_2_	1 × 10^3^ to 2 × 10^6^ nmol L^−1^	1 × 10^3^ nmol L^−1^	Wine	87.1% to 116.8%	[174]
ZnO NPs	SO_2_	5 × 10^3^ to 3 × 10^5^ ng mL^−1^	2 × 10^3^ ng mL^−1^	Wine	-	[175]
MoO_3__−__X_ nanosheets	Rhodamine 6G	-	100 nmol L^−1^	-	-	[176]

**Table 5 molecules-27-00508-t005:** The comparison of four representative detection methods of PADs.

Detection Methods	The Main Nanomaterials	Advantages	Disadvantages	Limit of Detection	Ref.
EC PADs	AuNPs, graphene, MOF, multiple metallic NPs composites and functionalized NPs	High sensitivity, rapid response and easy miniaturization	Equipment-dependent, complicated operation, easy interfered with complex matrices	1 pg mL^−1^ level for AuNPs, 0.1 pg mL^−1^ level for graphene, 0.1 mmol L^−1^ level for MOF, 1 × 10^−3^ pg mL^−1^ level for multiple metallic NPs composites and 1 × 10^−2^ pg mL^−1^ level for functionalized NPs	[40,50,56,68,75]
Colorimetric PADs	Functionalized AuNPs and AgNPs, enzyme-like nanomaterials	Simplicity and convenience, readout with naked eye, rapidness, low cost and low sample consumption	Poor sensitivity, limited to qualitative or semi-quantitative analysis	10 pg mL^−1^ level for functionalized AuNPs, 100 pmol L^−1^ level for functionalized AgNPs, 1 pmol L^−1^ level for enzyme-like nanomaterials	[48,89,95]
Fluorometric PADs	NC, QD, CD, MOF	Low cost, easy operation	Reader-dependent	1 pmol L^−1^ level for NC, 1 pg mL^−1^ level for QD, 1 × 10^3^ pg mL^−1^ level for CD, 10 pg mL^−1^ level for MOF	[115,127,144,157]
SERS PADs	AuNPs and AgNPs	On-site label-free detection, offered fingerprint signatures, high sensitivity	Equipment-dependent	1 pg mL^−1^ level for AuNPs and AgNPs	[166,170]

## Data Availability

Not applicable.

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
