# Peer review of "Enhancement of the Detection Performance of Paper-Based Analytical Devices by Nanomaterials"

_molecules, 2022, doi:10.3390/molecules27020508_

Round 1

Reviewer 1 Report

In the manuscript molecules-1505538, the authors wrote a review about the enhancement of uPADs’ performance using nanomaterials.

The review is up to date, and I would suggest its acceptance after major revision. My comments are:

  1. The format of writing "authors' name et al" reported/found... is unacceptable in the journal.
  2. Add “recovery” as a new column to the tables.
  3. ppb/ppm unit should write as ug L-1/mg L-1
  4. The format of writing "authors' name et al" reported/found... is unacceptable in the journal.
  5. Adding relevant figures of the reviewed devices will enrich the manuscript.
  6. A comparison table between these detection techniques (electrochemistry, colorimetry,…) is required.

Author Response

Thank you very much for your carefully review and constructive suggestions. We corrected the manuscript according to your suggestions, the changes/emendations have been highlighted by red color in the Marked-up Manuscript. The responses to your comments are as follows.

1) The format of writing "authors' name et al" reported/found... is unacceptable in the journal.

Response: Thank you very much for your comments.

We have changed "authors' name et al" to "authors' name et al." in the revised manuscript.

2) Add “recovery” as a new column to the tables.

Thank you very much for your suggestion.

According to your suggestion, a new column of“recovery” have added to the tables in the revised manuscript.

3) ppb/ppm unit should write as ug L-1/mg L-1

Thank you very much for your advice.

According to your advice, we have changed ppb/ppm to ng mL-1 or μg mL-1 in the revised manuscript.

4) Adding relevant figures of the reviewed devices will enrich the manuscript.

Thank you very much for your suggestion.

The relevant schematic diagrams of the reviewed devices have been added in the revised manuscript (see Figure 3 to 6).

5) A comparison table between these detection techniques (electrochemistry, colorimetry,…) is required.

Thank you very much for your suggestion.

We have added a comparison table between these detection techniques, shown as Table 5 in the revised manuscript.

Reviewer 2 Report

Comments are added to the pdf file

Author Response

Response: Thank you very much for your carefully review and constructive suggestions. We corrected the manuscript according to your suggestions, the changes/emendations have been highlighted by red color in the Marked-up Manuscript. The responses to your comments are as follows.

1) ‘this is confusing. 2021 is not over. I believe the whole graph would look totally different if you separate this to two figures a and b with some "cutting" of the scale’ in Figure 1.

Thank you very much for your advice.

Although 2021 is not over yet, we have marked the date range in figure 1 as ‘from 01-01-2015 to 30-10-2021’. According to your advice, we separated Figure 1 to two figures with some "cutting" of the scale in the revised manuscript. Generally, the publication numbers in the right figure are less than those in the left figure because the nanomaterials are not used in some publications.

2) Add references behind the sentence of ‘Several AuNPs labeled lateral-flow test-strip (LFTS) immunosensors such as human chorionic gonadotropin (HCG) and Hepatitis B surface antigen (HBsAg) colloidal gold immunoassay strips, have been clinically approved for rapid testing’.

Thank you very much for your advice.

The human chorionic gonadotropin (HCG) and Hepatitis B surface antigen (HBsAg) colloidal gold immunoassay strips have extensively used around the world. We believe that readers can get relevant information from manufacturers such as Abbott, Church & Dwight, Prestige Brands, Quidel, Boots Pharmaceuticals, Confirm BioSciences, Germaine Laboratories, KIP Diagnostics, Map Diagnostics, Piramal Healthcare, Philippine Blue Cross Biotech, Princeton BioMeditech, Rite-Aid, Mankind Pharma, etc..

3) ‘that's already mentioned in the box on the right’ in Figure 2.

Thank you very much for your comments.

According to your suggestion, we deleted ‘Electrochemical μPAD (ePAD)’ in the middle column of the box, and changed Figure 2 to a new one in the revised manuscript.

4) ‘scheme of how do they work’ about the nanomaterial-enhanced paper-based analytical device in part 2.

Thank you very much for your suggestion.

According to your suggestion, we added Figure 3 to 6 to illustrate the work process of ePAD, colorimetric, fluorometric and SERS PAD respectively in the revised manuscript.

5) ‘what is the order in here? nanomaterial? modification method?’ in Table 1 and 2.

Thank you very much for your question.

We are sorry for the confusing order in the tables, we reordered the tables by nanomaterials, and updated them in the revised manuscript.

6) ‘if possible recalculate to the same units everywhere so one could compare it’ in the tables.

Thank you very much for your suggestion.

According to your suggestion, we unified the units to ng mL-1 or nmol L-1 in the revised manuscript.

7) ‘order of references in this table’ in Table 4.

Thank you very much for your advice.

Round 2

Reviewer 1 Report

The authors have improved the manuscript and responded adequately to all my questions. Therefore, I recommend that this manuscript be accepted.

Reviewer 2 Report

double check for typos